# To Consent or Not to Consent to Screening, That Is the Question

**DOI:** 10.3390/healthcare11070982

**Published:** 2023-03-30

**Authors:** Bjørn Hofmann

**Affiliations:** 1Centre for Medical Ethics, Faculty of Medicine, University of Oslo, P.O. Box 1130, N-0318 Oslo, Norway; b.m.hofmann@medisin.uio.no; 2Department of Health Sciences, Norwegian University of Science and Technology, P.O. Box 191, N-2802 Gjøvik, Norway

**Keywords:** consent, autonomy, ethical rationalities, bias, information, disclosure

## Abstract

The objective of this article is to address the controversial question of whether consent is relevant for persons invited to participate in screening programs. To do so, it starts by presenting a case where the provided information historically has not been sufficient for obtaining valid informed consent for screening. Then, the article investigates some of the most relevant biases that cast doubt on the potential for satisfying standard criteria for informed consent. This may indicate that both in theory and in practice, it can be difficult to obtain valid consent for screening programs. Such an inference is profoundly worrisome, as invitees to screening programs are healthy individuals most suited to make autonomous decisions. Thus, if consent is not relevant for screening, it may not be relevant for a wide range of other health services. As such, the lack of valid consent in screening raises the question of the relevance of one of the basic ethical principles in healthcare (respect for autonomy), one of the most prominent legal norms in health legislation (informed consent), and one of the most basic tenets of liberal democracies (individual autonomy). Thus, there are good reasons to provide open, transparent, and balanced information and minimize biases in order to ascertain informed consent in screening.

## 1. Introduction

Informed consent for screening has been a crucial issue since the conception of the various screening programs. As screening was conceived of as a health good, high uptake has been a primary goal both in initial research trials and in the implementation and provision of screening programs [1,2,3,4]. Accordingly, screening programs have been accused of overselling benefits [5,6], of not providing balanced information to the invited [7,8,9,10,11,12,13,14,15,16], and of nudging invitees toward participation [17,18]. 

Systematic reviews have documented that both the general population and health professionals have biased expectations of the benefits and harms of health screening programs [19,20]. Moreover, a recent literature review identified six major categories of systematic influence of invitees’ decisions: misleading presentation of statistics, misrepresentation of harms vs. benefits, opt-out systems, recommendation of participation, fear appeals, and influencing general practitioners and other healthcare professionals [21].

Hence, the implementation and provision of screening programs has raised the ethical question of whether real informed consent is obtained.

This is paradoxical as there are at least two good reasons why informed consent is relevant in screening. First, screening programs target healthy persons who are not in pain or despair. Hence, their ability to understand information and to deliberate based on this information is as good as can be. Second, screening programs are not perfect. They come with potential benefits (reduced morbidity and mortality) and harms (overdiagnosis, overtreatment, anxiety). Moreover, screening programs have been controversial especially with respect to their safety, effectiveness, and efficiency [22,23].

In liberal democracies, the trade-off between benefits and harms of individual health deliberations are expected to be made by the individual. However, disclosed information is only partly understood [24,25,26] and decisions on participating in screening programs are only partly (or not at all) taken on the basis of information of potential benefits and risks [27,28,29,30,31,32]. A range of other (social) factors are at play, such as risk perception, peer attitudes, conformity (with peers and health authorities), routinization, and trust [3,33]. While information about screening programs has improved significantly, this does not seem to have improved the consent process.

Hence, if understanding relevant information is a precondition for valid informed consent [34], then valid consent may be rare. This raises the provocative *research question of whether consent is relevant for participating in screening programs* at all. Accordingly, the objective of this article to investigate whether consent is relevant for persons invited to participate in screening programs.

To address this question, this article starts by analyzing the information provided to Norwegian women invited to the national Mammography screening program from 1996 to 2021. This illustrates that basic preconditions for informed consent have not been fulfilled. Thereafter, the article analyzes other factors undermining informed consent as required in the ethics literature. In particular, it investigates various biases and social mechanisms.

While it can be argued that the biases make information efforts obsolete [35,36], I argue on the contrary that the many biases make it even more important to provide balanced quality assured information and encouragement to the individual to make up their own mind. If I am wrong, this has the very unpleasant implication that consent is not relevant for screening programs. Even more, it is not relevant for other healthcare interventions either, where patients may be in pain and despair and much less able to receive and understand information and deliberate on it.

For the analysis, I apply a standard definition of consent: “A person gives an informed consent to an intervention if he or she is competent to act, receives a thorough disclosure, comprehends the disclosure, acts voluntarily, and consents to the intervention” [34]. Disclosure refers to information necessary for deliberation and decision making. Intervention includes the practical parts of screening tests.

## 2. Poor Information Undermines Informed Consent

The disclosure of information is a prerequisite for understanding, which in turn is a requirement for informed consent. I will apply a case study of information from the Norwegian Mammography Screening Program (NMSP) as a case to illustrate a development which is typical for a range of screening programs.

Many screening trials and subsequent screening programs have been accused of not informing sufficiently about the potential harms and benefits [3,11,15,37,38,39]. As a result, many of them have improved their information to the public, which is frequently assisted by external or independent researchers. The NMSP is but one example of this; see Table 1. While the program has improved its information—in part based on explicit critique [40]—it is still mainly based on its own data analysis and not on analyses from an external independent assessment by the Norwegian Research Council [41].

It is important to notice that the evidence from clinical trials has evolved over the years and that more information has become available. However, although information has been available from research, this has not been conveyed in the invitations or information material, and even when information has been available from independent sources, such as from the NRC, this has not been conveyed or has been supplemented with information from research completed by the NMSP itself.

Hence, one reason why it has been and still can be difficult to obtain real informed consent for participating in screening programs is that the information is not provided in an open, transparent, or balanced manner based on independent sources. As pointed out in a systematic review and meta-analysis, “[i]nformation on cancer screening is often biased, incomplete and persuasive” [42]. See also [25,43,44]. Moreover, using balanced decision aids has shown pro-screening decisions to decrease [42].

Hence, one of the basic preconditions for informed consent, open, transparent, and balanced information, has not been available, thus undermining real informed consent. Yet another reason why real informed consent is not obtained is that participants do not want to know the detailed information.

### 2.1. Information Is of Minor Importance for Decisions to Participate in Screening Programs

As a wide range of studies have shown, information from health providers or health authorities about screening program potential benefits and harms are of minor importance for invitees’ decisions to participate [27,28,29,30,31,32]. Other factors tend to be more important, such as trust in the provider (or health authority), symbolic value (e.g., for women’s health in breast cancer screening), fear, because it is offered (by healthcare systems or it is for free), and so-called rutinization [44,45].

Hence, information may not be as important for deliberation as the ideal model of consent may presume. One reason for this may be a series of psychological biases.

### 2.2. Biases in Screening Decisions

A wide range of cognitive and affective biases have been identified in psychology and behavioral economics [38,46,47,48,49,50,51], many of which are relevant for decisions on participating in screening [52,53,54].

As already mentioned, expectations with respect to benefits (overestimated) and harms (underestimated) are biased both amongst the general population and health professionals [19,20].

It has also been shown that people are subject to *anchoring-and-adjustment bias*; i.e., they insufficiently adjust their subjective risk to the objective risk value communicated to them [55]. This may undermine the comprehension criteria of informed consent.

People also have a tendency to rely on emotions, rather than concrete information, when making decisions. This has been called *affect heuristic*. Emotions that are not founded in evidence can lead to unjustified decisions in screening [56]. As can be seen by several of the quotations in Table 1, there is a strong appeal to affect heuristic, e.g., “Think about your future—take advantage of the offer of mammograms!” and “A mammogram can save lives”.

*Ambiguity aversion*, which is also called uncertainty aversion, is characterized as the tendency of people to have a preference for risks that are known over those that are unknown and has been studied in the setting of screening [57,58,59]. As demonstrated in the case above, there have been and still are many unknown risks in screening, and not taking all risks into account may undermine informed consent.

Moreover, people may have a tendency to rely on immediate examples that are available or come to a given person’s mind when making decisions. There are many examples in the media of people who have been “saved by screening” or were “discovered too late” due to lack of screening, while there are few stories of people being overdiagnosed and overtreated. (See the “popularity paradox” below.) This *availability bias* makes decision-makers rely on unbalanced or incomprehensive information when making decisions on screening [60].

The *bandwagon effect* is the tendency for people to adopt certain attitudes or behaviors because others are doing so, i.e., to “follow the rest” or “group think” [61]. This can undermine informed consent because decisions are not based on comprehension of the decision or on own deliberation. From the Norwegian case, the following information is interesting: “Three out of four invited women choose to participate.” See Table 1.

Physicians, patients, and the general population appear to share a tendency toward action rather than inaction, i.e., *commission bias* [62]. This tendency may bias invitees to accept invitations and attend screening programs. Commission bias is related to other cognitive biases, such as the tendency to think that it is better to know than not to know and that early detection is better than late detection [63,64]. “Regular mammography is today the most important the method of detecting breast cancer at an early stage” (Table 1).

People have been shown to have a tendency to interpret new information and evidence as confirmation of existing beliefs, conceptions, or theories. This has been called *confirmation bias*, and it can reduce critical assessment of the evidence and result in reduced comprehension and biased decisions [65]. “Early diagnosis, easier treatment, better life expectancy.” (Table 1)

As high uptake (participation rates) has been a goal in various screening programs, the *decoy effect* is relevant for screening. This effect can be obtained by increasing the interest in a target action (participation) by introducing an inferior alternative choice (decoy) and has been demonstrated for colorectal cancer screening [66]. However, using decoys would be to lure people toward specific choices and would undermine deliberation and informed consent.

*Default bias* is the tendency to stay in or make the default choice, and it has been used as a strategy to influence screening participation, e.g., through opt-out systems [21]. The Norwegian system with fixed invitations and specific appointments is but one example of this. Providing a default choice undermines real informed choice and thus informed consent. In the Norwegian case, the following may serve as an example: “We hope you choose to participate in the Public Mammography Program” (Table 1). Moreover, offering a public screening program (for free) is considered as a recommendation to participate and invitees “perceive the choice of opting into the screening program as already made, while a decision to opt out would have demanded more effort” [34]. Accordingly, participation can become subject to routinization [67].

The tendency for people to decide based on how the information is presented (framed) is called *the framing effect*. This bias has been demonstrated in a wide range of health-related decision making, including screening [68,69]. For example, whether outcomes are presented in frequencies or percentages, as relative or absolute risk, or as losses or gains can influence choice [70]. The challenge with framed information is that it can reduce the ability to comprehend and deliberate on information and thus to give valid consent.

*Impact bias* (affective forecasting) is the tendency for people to overestimate the impact that future events will have on their lives. For example, invitees can overestimate the risk of cancer or decide to participate because they are afraid of the affective burden of regretting (anticipated decision regret) [71]. The problem with these variants of bias is that they can undermine valid informed consent [72,73].

The tendency for people to underestimate their probability of experiencing adverse effects is also relevant for screening. This *optimism bias* can be seen in what has been called “the popularity paradox” where people who have been overdiagnosed and overtreated as a result of screening programs tend to think that they have been saved by screening [74,75]. Underestimating the risk of overdiagnosis and overtreatment can bias decision-making and undermine valid informed consent.

The way that information about screening is presented may also influence decision making, as people have a tendency to pay more attention to information presented first (and last). This has been called *the order effects*. Unbalanced attention to information may bias decision making and informed consent.

*Representativeness heuristic* is the tendency to base present decisions on past events or experiences that appear similar to the current situation. Many people decide to participate in screening because they know of persons who have died of the disease screened for or who have been identified as diseased due to screening. Decisions based on knowledge of persons with specific diseases or screening experiences rather than own relevant risk assessments may be biased and undermine valid informed consent.

These are but some of the very many types of biases that can undermine consent in decisions to participate in medical screening, and a summary is provided in Table 2.

## 3. Discussion

In this article, I have presented two issues that may undermine the relevance of consent for persons invited to screening programs. The first is a historical and practical problem, as information from screening programs has not been open, transparent, and balanced. I have used the information provided in the Norwegian mammography screening program as a case to discuss the general challenge of obtaining valid informed consent for screening. While the provided information clearly has improved in quality and quantity of information disclosure, it still casts doubt on whether it satisfies standard criteria for informed consent. Moreover, invitees may not want balanced information. The second issue undermining consent is a wide range of biases that distort the information understanding, the deliberation process, and the decision making. Such biases can explain both the problems with obtaining informed consent (seen in the case), and they challenge the institution of informed consent as such. That is, it raises the question of whether we should have consent for screening programs in the first place.

One argument against consent is that preventive medicine in general and screening programs in general are perceived as paternalistic by the general population, especially in trust-based healthcare systems [3]. Accordingly, informed consent is neither desirable nor necessary.

Others have argued that not only are screening decisions unfree and uninformed, but they should be nudged as well. For an overview of the various arguments for nudging, see [36]. Moreover, others have argued for alternative forms of consent, such as broad consent [76], which is widely used in biobank research.

However, if health services offered to healthy persons who are well equipped to do so should not consent, then one can question the role of consent in general. If persons who have the capacity to consent, to understand the provided information, to deliberate on disclosed information, and to make voluntary choices should not consent to a health services, such as screening, one may legitimately question the relevance of informed consent and its justifying principle of respect for autonomy in general.

Hence, the lack of valid consent in screening raises the question of one of the basic ethical principles in healthcare (autonomy), one of the most prominent legal norms in health legislation (informed consent), and one of the most basic tenets of liberal democracies. This makes the list of biases and the challenges with balanced disclosure illustrated in the case from Norway worrisome, especially as most screening programs work under the assumption and requirement of informed consent.

This also provides the most profound argument for striving to obtain real consent in screening (and in other types of healthcare provision): sustaining liberal democracy.

### 3.1. Limitations

There are certainly many limitations to this article. Firstly, the case study is limited to only one country, Norway, which obviously may be an outlier. However, as can be seen from the referred literature, the challenges with open, transparent, and balanced information are ubiquitous. Hence, while other countries very well may have improved more than Norway, the problem of information from screening programs appears relevant.

Secondly, there are very many biases that are relevant for decision making with respect to screening programs that have not been included. Moreover, this article has been limited to the invitees’ decision making. There are many biases for health professionals and health policy makers as well. These enhance the biases of the invitees. For example, the public health interest in increasing participation in screening programs enhances the framing effect and default bias.

Many bias-related issues have not been addressed either, such as imperatives, conflict of interest, and polarized research [77]. Hence, the article is not exhaustive. However, exhaustiveness is not necessary for the main argument: undermining consent in screening has vast ethical, legal, and societal implications.

One relevant counterargument to this is that according to the definition of consent, disclosed information should be adapted to the person’s preferences (according to a subjective standard). Disclosure is defined as informing about “(1) those facts or descriptions that patients or subjects consider material when deciding whether to refuse or consent to a proposed intervention or involvement in research, (2) information the professional believes to be material, (3) the professional’s recommendation (if any), (4) the purpose of seeking consent, and (5) the nature and limits of consent as an act of authorization.” According to the first requirement, one can argue that when people do not consider detailed information about screening programs to be material to their participation, they can consent without being informed about these issues. The problem with this is of course that in order to know whether information is material or not (and the nature and limits of consent), you need to have at least some minimal knowledge about the content of the intervention, such as the risk and benefits.

There are also a range of medico-legal aspects with respect to autonomy and informed consent that support the main point of this article [78]. This is the topic of a separate study and is beyond the scope of this article.

### 3.2. What Should We Do?

Before abandoning consent for screening programs (and other health services), we should consider alternatives. The following suggestions has been made: “forewarning patients about the bias, tailoring risk information to their numeracy level, emphasizing social roles, increasing motivation to form accurate risk perception, and reducing social stigmatization, disease worry and information overload” [55]. Other measures have also been suggested, such as debiasing [79,80,81,82], decision support [83,84,85] as well as interventions to improve decision-making skills [86]. Whether such measures will do the trick is still open for discussion.

Even more, shared decision making (SDM) may also be an interactive mode of communication that could reduce the effect of some of the biases. However, SDM has its own challenges that have to be taken into account [87].

While some would argue that we therefore should abandon consent for screening, this would generate precedence for a wide range of other health services. Moreover, it would undermine the basis for deliberative democracy. Yet another alternative would be to abandon screening programs where valid consent cannot be obtained. This has been suggested for breast cancer screening (6). While reasonable, it may not be feasible. Hence, we should strive for valid consent in screening by providing open, transparent, and balanced information and promote unbiased deliberation.

## 4. Conclusions

This article has raised the question of whether consent is relevant for persons invited to screening programs. The information provided in the Norwegian mammography screening program has been used as a case to discuss the general challenge of obtaining valid informed consent for screening. In addition to the history of unbalanced information, a review of some of the most relevant biases casts doubt on the potential for satisfying standard criteria for informed consent. This is worrisome, as invitees to screening programs are healthy individuals most suited to make autonomous decisions. Thus, if consent is not relevant for screening, it may not be relevant for a wide range of other health services. As such, the lack of valid consent in screening raises the question of one of the basic ethical principles in healthcare, which is one of the most prominent legal norms in health legislation and one of the most basic tenets of liberal democracies. Thus, there are good reasons to strive for open, transparent, and balanced information and minimize biases in order to ascertain informed consent in screening.

## Figures and Tables

**Table 1 healthcare-11-00982-t001:** Content elements of the Norwegian Mammography Screening Program (NMSP) by year *.

Topic\Year	1996	2003	2009	2017	2021	Independent Report, NRC, 2015
Reduced risk of death (overall mortality reduction)	No information provided	No information provided	No information provided	No information provided	No information provided	No information provided
Reduction in breast cancer mortality (breast cancer-specific mortality reduction)	“Mortality of breast cancer can be reduced by about one-third at systematic health examinations with mammography.”	“Regular participation in the mammography program reduces the risk of dying from breast cancer.”	“Regular mammography reduces breast cancer mortality.” “Annually about 1000 cases of breast cancer or precancerous breast cancer are detected” by the program.	“6 [out of 1000] women are diagnosed with breast cancer that needs to be treated” (in figure) “The main benefit of mammography screening is that it leads to fewer deaths from breast cancer among women in the target group.”	“If 1000 women attend all invitations in the Mammography Program, about 4 of them will avoid dying of breast cancer as a result of the program.” “The calculation from the public evaluation indicates that if 1000 women in their 50s are invited to the mammography program ten times, it can be expected that about 3 of them avoid dying from breast cancer.”	20–30% for women between 50 and 70 years. 27 out of 10,000 women, 50 years old, screened for 10 years, who attended 10 times and with an attendance rate of 76%.
False positive test result	No information provided	No information provided	“About 4 out of 100 who participate are recalled for a more thorough examination.” “For most people, it turns out that the changes are harmless, and this is referred to as a false positive mammography examination.”	24 out of 1000This is not explicitly stated but must be calculated: 30−6 = 24. However, it states that: “18 women need new mammograms and/or ultrasound and are then told that there are no malignant findings.”	200 of 1000 screened ten times. 160 of these will have new mammograms and/or ultrasound examinations.Of these 40 will need biopsies.	20% when participating in all examinations for 10 years.1520 out of 10,000 followed for 10 years.
Overdiagnosis	No information provided	No information provided	No numbers. “As of today, it is not possible to predict how or how quickly a screening-detected precancerous condition or case of breast cancer will develop.” “There is disagreement in the academic community about how big the problem is.”	No estimates are given. “Mammography screening will entail a risk of overdiagnosis *. At present, it is not possible to distinguish which cancer cases are overdiagnosed, and therefore, everyone with detected breast cancer is offered treatment»	“if 1000 women attend all the times they are invited by the Mammography Program, about 12 of them will be overdiagnosed as a result of the program.”	142 out of 10,000 15–25% for women between 50–79 years compared to those who do not receive an invitation. 15–20% for the same age group if invited.
Overtreatment	Not mentioned	Not mentioned	Not mentioned	Not mentioned	Mentioned	Mentioned but not estimated
Interval cancer (cancer that occurred in the period between attending for mammography at the program)	Not mentioned, but women are encouraged to examine the breasts themselves.	“Mammography does not reveal all changes in the breasts.”	“Some cases of breast cancer are not detected by mammography or occur in the time between two mammography examinations.”	2 out of 1000 “2 women [out of 1000 will] be diagnosed with breast cancer in the time before the next survey.”	2 of 1000 1 of 4 cancer cases in the program “If 1000 women attend the program, 2 of them will experience breast cancer detection between two screening examinations”	25% of all participants in screening. 127 out of 10,000 followed for 10 years. Also informs about 42 false negative responses out of 10,000
Anxiety, uneasiness	Not mentioned	“An invitation for a post-examination may cause anxiety, but as a participant in the public program, you are ensured prompt follow-up.”	“Many people may experience anxiety and uneasiness in connection with the mammography examination, both in the time leading up to the results and when summoning additional examinations.”	“In connection with the mammography examination one can experience anxiety and uneasiness, both in the time leading up to the answer and when summoning for an additional examination.”	“Uneasiness and anxiety is a common reaction when waiting for an answer”	Mentioned, but not quantified
Recalls	1 of 20, 5 of 100	3 of 100	4 of 100	3 of 100	160 of 1000	1520 of 10,000. 20% after 10 invitations.
Number of treatment-demanding findings	No information provided	6 of 1000	5–6 of 1000	6 of 1000	76 of 1000 76 are diagnosed and ”all women with detected breast cancer will be offered treatment”	
Potentially biased content	”Think about your future-take advantage of the offer of mammograms!” “You should take advantage of this offer.” “A mammogram can save lives.”	“A mammogram can save lives.” “Early diagnosis, easier treatment, better life expectancy.” “Participation inmammography program ensures fast follow-up.” “We hope you choose to participate in the public mammography program.”	“Three out of four invited women choose to participate.” “Regular mammography is today the most important method of detecting breast cancer at an early stage.”	“We really appreciate letting us know if you don’t come, then others can enjoy your appointment.” “Do you want to participate in the mammography program?”	Favoring own estimates to the independent evaluation (NRC report)	NA

* The material stems from the publicly available information from the NMSP as well as in the independent report by the Norwegian Research Council (NRC). For details, see [40]. The content is translated from Norwegian by the author.

**Table 2 healthcare-11-00982-t002:** A summary of relevant biases (and heuristics) for decision making on participating in screening programs. Useful reviews of the general literature can be found in [52,53].

Bias	Description	Relevance for Consent in Screening
Affect heuristic	The tendency to rely on emotions, rather than concrete information, when making decisions	Emotions not founded in evidence may lead to unjustified decisions
Ambiguity aversion (uncertainty aversion)	A preference for known risks over unknown risks	There are many unknown risks in screening
Anchoring bias	The tendency to insufficiently adjust subjective risk to the objective risk value communicated to people	Conceptions about the risks and benefits of participating in screening is not modified by factual information
Availability bias	The tendency to rely on immediate examples that come to a given person’s mind when making decisions	Information applied in decisions may be anecdotal, unbalanced, or incomprehensive
Bandwagon effect	The tendency for people to adopt certain behaviors because others are doing so	Decisions are not based on comprehension or on own deliberation
Commission bias	The tendency toward action rather than inaction	Biases decisions toward accepting invitations
Confirmation bias	The tendency to interpret new information as confirmation of existing beliefs, conceptions, or theories	Interpreting new information as confirmation of existing beliefs may reduce critical assessment of the evidence and result in biased decisions
Decoy effect	Increasing the interest in a target action inclusion by introducing an inferior alternative choice (decoy)	Using decoys would be to lure people toward specific choices and would undermine deliberation
Default bias	The tendency to stay in or make the default choice	Providing a default choice undermines real informed choice
Framing effect	The tendency for people to decide based on how the information is presented (framed)	Framed information reduces the ability to comprehend and deliberate on information
Impact bias (Affective forecasting)	The tendency for people to overestimate the impact that future events will have on their lives	Overestimating the risk of cancer can bias decision making
Optimism bias	The tendency for people to underestimate their probability of experiencing adverse effects	Underestimating the risk of overdiagnosis and overtreatment can bias decision making
Order effects: primacy/recency	The tendency to pay more attention to information presented first (and last)	Unbalanced attention to information may bias decision making
Representativeness heuristic	The tendency to base present decisions on past events or experiences that appear similar to the current situation	Decisions can be based on knowledge of persons having screening experiences rather than own relevant risk assessments

## Data Availability

All data available in publication.

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
