# Peer review of "To Consent or Not to Consent to Screening, That Is the Question"

_healthcare, 2023, doi:10.3390/healthcare11070982_

Round 1

Reviewer 1 Report

The article presents an interesting case for informed consent in screening programs. What I would suggest is that the author makes it very clear at the beginning that informed consent is necessary for screening programs. Another suggestion is that the author should present his argument in a more substantive manner. They should present the main argument in a clear format - why informed consent is necessary for screening programs. One thing is that competent and healthy persons have no difficulty in providing the consent, but this is not exactly an argument for why informed consent is necessary. The fact that they can do it does not mean that they must do it. Another argument is that giving consent is in line with liberal democracy. The author should present some reason and argument to show why the two - democracy and content - are linked. This is missing in the paper.

Author Response

RESPONSE:  I am most thankful for these helpful comments and suggestions, which I have addressed in the following manner: I now make it clear that consent is crucial for screening programs in the abstract, introduction, in the discussion, as well as in the conclusion. In particular, I write that “As such, the lack of valid consent in screening raises the question of the relevance of one of the basic ethical principles in healthcare (respect for autonomy), one of the most prominent legal norms in health legislation (informed consent), and one of the most basic tenets of liberal democracies (individual autonomy). Thus, there are good reasons to provide open, transparent, and balanced information and minimize biases in order to ascertain informed consent in screening.”

Moreover, the argument for consent is sharpened. For example, the following has been added: “Hence, the lack of valid consent in screening raises the question of one of the basic ethical principles in healthcare (autonomy), one of the most prominent legal norms in health legislation (informed consent), and one of the most basic tenets of liberal democracies. This makes the list of biases and the challenges with balanced disclosure illustrated in the case from Norway worrisome. Especially as most screening programs work under the assumption and requirement of informed consent. This also provides the most profound argument for striving to obtain real consent in screening (and in other types of healthcare provision): sustaining liberal democracy.”

In addition, several other improvements have been done in according to the comments and suggestions from the other reviewers.

Reviewer 2 Report

Thank you for an interesting paper. I wonder if it might have been better to use the latest version of the mammography screening information and contrast this with the independent report? I found the table with historic versions difficult to read and it did not really substantiate your argument as there were so many different versions.

I also wonder about the place for shared decision-making in this arena, as an alternative to informed consent? Perhaps you could introduce this topic and its requirements, and suggest it as an appropriate strategy which could remove sole reliance on printed information?

Otherwise I enjoyed your argument for the importance of consent in these patients.

I have attached a file with suggestions for some minor edits.

Author Response

REESPONSE: I am most thankful for the good comments and constructive suggestions. All the comments in the attached file have been followed. I am also very grateful for suggesting to include shared decision-making. This now reads: “Even more, shared decision-making (SDM) may also be an interactive mode of communication that could reduce the effect of some of the biases. However, SDM has its own challenges that have to be taken into account (87).” I also agree that Table 1 is comprehensive, and thus can be confusing. Using the independent report to contrast the latest version of the information is a good suggestion. Unfortunately, this is difficult as the report is not produced for informing invitees in the same manner as the information from the Mammography screening program. Even more, selecting one of the version of the information from the Program would be considered to be unfair by the Program and would conceal the historical development.

In addition, several other improvements have been done in according to the comments and suggestions from the other reviewers.

Reviewer 3 Report

I read the article with attention and interest. I believe that the topic is certainly topical and useful for international readers and for this reason I think it should be published, however I find some suggestions that may be useful.

- the method does not collect particular data, so I think it is better to classify it as a commentary or opinion and not as an "article"

- there are many refereces of the author, I think he should give up some.

- I think it is also useful to refer to what could be the medico-legal problems due to a lack of consent, for example: doi: 10.1111/vox.13106. Epub 2021 Apr 7

- the author could then make a constructive proposal regarding the possibility of a better consensus, how would he articulate it?

Author Response

REESPONSE: I am most thankful for these helpful comments and suggestions. In many ways this is an analysis article. However, it contains the results from an analysis of information material (data) from the Norwegian Mammography Screening Program. I will leave it to the Editor to decide the article category. Please also note that this manuscript is a contribution for a special issue on «The rationalities of medical screening» https://www.mdpi.com/journal/healthcare/special_issues/Medical_Screening

I would also reduce the number of references to myself on the Editor’s advice. Preferably not those references on the analysis of the Norwegian information on mammography screening.

The issue of medico-legal issues is well taken. The following has been added to the discussion: “There are also a range of medico-legal aspects with respect to autonomy and informed consent that support the main point of this article (79). As this is the topic of a separate study and are beyond the scope of this article.”

With respect to a more constructive proposal, several paragraphs have been added to the section “What should we do,” including the following: “While some would argue that we therefore should abandon consent for screening, this would generate precedence for a wide range of other health services. Moreover, it would undermine the basis for deliberative democracy. Yet another alternative would be to abandon screening programs where valid consent cannot be obtained. This has been suggested for breast cancer screening (6). While reasonable, it may not be feasible. Hence, we should strive for valid consent in screening by providing open, transparent, and balanced information and promote unbiased deliberation.” Moreover, the following has been added to the Discussion: “Hence, the lack of valid consent in screening raises the question of one of the basic ethical principles in healthcare (autonomy), one of the most prominent legal norms in health legislation (informed consent), and one of the most basic tenets of liberal democracies. This makes the list of biases and the challenges with balanced disclosure illustrated in the case from Norway worrisome. Especially as most screening programs work under the assumption and requirement of informed consent. This also provides the most profound argument for striving to obtain real consent in screening (and in other types of healthcare provision): sustaining liberal democracy.”

In addition, several other improvements have been done in according to the comments and suggestions from the other reviewers.

Reviewer 4 Report

Please find in the following my comments about the review of a manuscript under the title (To Consent or Not to Consent to Screening, That Is the Question). This article has raised the question of whether consent is relevant for persons invited to screening programs.

Originality and relevance

§  The study is interesting for reading.

§  The study has moderate scientific quality.

§  The study is relevant to the scope of this journal.

§  The manuscript is relevant to the field and its presentation needs minor modifications to be clearer.

Few points in the study need to be added and changed to adhere to the journal's standards. If the authors will revise with proper objectives, organization and supportive evidence, hypothesis, and their claim clearly, it may be considered for publication

Comments:

Abstract:

§  The aim of the study should be included in the abstract.

§  Add more details about the methodology and mentioned the most prominent finding in the results.

Introduction

§ rationale of the study is not clear in the introduction section.

§  The procedure for the study should be described in detail.

Poor information undermines informed consent.

§  Table 1 is better to be divided into two tables.

§  In the title of table one, the authors mentioned (For details, see (14). 79 Translations by the author), clarify this point.

Discussion:

§  The findings need more interpretations.

§  Add a section under the title (Limitation of study).

Language

§  Language and editing should be revised all over the manuscript.

Author Response

I am most thankful for these helpful comments and suggestions, which I have addressed in the following manner:

Abstract: The abstract has been rewritten in accordance with the suggestion.

Introduction: The rationale and procedure has been clarified. In particular, a section has been elaborated as follows: “Hence, if understanding relevant information is a precondition for valid informed consent (35), then valid consent may be rare. This raises the provocative question of whether consent is relevant for participating in screening programs at all. Accordingly, the objective of this study is to investigate whether consent is relevant for persons invited to participate in screening programs.

To address this question, this study starts by analysing the information to Norwegian women invited to the Mammography screening program. This illustrates that basic preconditions for informed consent are not fulfilled. Thereafter, the article analyses other factors undermining informed consent as required in the ethics literature. In particular, it investigates various biases and social mechanisms.”

I would be happy to divide Table 1 in two, as suggested. However, I would need some advice on how to do so. Should it be done according to the content of rows or temporal (columns)? Would you prefer to delete some of the columns? In any way, the legend has been revised.

More interpretations are provided, and two sections have been added under the title (Limitation of study) as suggested.

The language has been revised.

In addition, several other improvements have been done in according to the comments and suggestions from the other reviewers.

Round 2

Reviewer 3 Report

I appreciated the improvements indicated. I think the article can be published

Author Response

I am most thankful for the positive appraisal of the revision.